# Comparison of biosimilar Tigerase and Pulmozyme in long-term symptomatic therapy of patients with cystic fibrosis and severe pulmonary impairment (subgroup analysis of a Phase III randomized open-label clinical trial (NCT04468100))

Elena L. Amelina[1], Stanislav A. Krasovsky[1], Nina E. Akhtyamova-Givirovskaya[2]*, Nataliya Yu. Kashirskaya[3], Diana I. Abdulganieva[4], Irina K. Asherova[5], Ilya E. Zilber[6], Liliya S. Kozyreva[7], Lubov M. Kudelya[8], Natalya D. Ponomareva[9], Nataliya P. Revel-Muroz[10], Elena M. Reutskaya[11], Tatiana A. Stepanenko[12], Gulnara N. Seitova[13], Olga P. Ukhanova[14], Olga V. Magnitskaya[15], Dmitry A. Kudlay[2], Oksana A. Markova[2], Elena V. Gapchenko[2]

**1** State Budgetary Healthcare Facility of Moscow City D.D. Pletnyov City Clinical Hospital of the Department of Healthcare of Moscow, Moscow, Russia, **2** JSC GENERIUM, Moscow, Russia, **3** Research Centre for Medical Genetics, Moscow, Russia, **4** Department of Hospital Therapy, Federal State Budgetary Educational Institution of Higher Education Kazan State Medical University of the Ministry of Health of the Russian Federation, Kazan, Russia, **5** State Healthcare Institution of Yaroslavl Region Children's Clinical Hospital No. 1, Cystic Fibrosis Centre, Yaroslavl, Russia, **6** State Autonomous Healthcare Institution of Yaroslavl Region Clinical Hospital No. 2, Yaroslavl, Russia, **7** State Budgetary Healthcare Institution G.G. Kuvatov Republican Clinical Hospital, Republic of Bashkortostan, Russia, **8** State Budgetary Healthcare Institution of Novosibirsk Region State Novosibirsk Regional Clinical Hospital, Novosibirsk, Russia, **9** State Budgetary Healthcare Institution of Sverdlovsk Region Sverdlovsk Regional Clinical Hospital No. 1, Yekaterinburg, Russia, **10** State Budgetary Healthcare Institution Chelyabinsk Regional Clinical Hospital, Chelyabinsk, Russia, **11** Krai State Budgetary Healthcare Institution Krai Clinical Hospital (KSBHI Krai Clinical Hospital), Barnaul, Russia, **12** State Budgetary Healthcare Institution Municipal Multidisciplinary Hospital No. 2, Saint Petersburg, Russia, **13** Federal State Budgetary Scientific Institution Tomsk National Research Medical Centre of the Russian Academy of Sciences (Tomsk NRMC), Clinic of Genetics, Research Scientific Institute of Medical Genetics, Tomsk, Russia, **14** Research Medical Centre for General Therapy and Pharmacology, Limited Liability Company, Stavropol, Russia, **15** Department of Clinical Pharmacology and Intensive Care, Volgograd State Medical University, Volgograd, Russia

* neakhtyamova-givirovskaya@generium.ru

## Abstract

### Background

Patients with cystic fibrosis (CF) need costly medical care and adequate therapy with expensive medicinal products. Tigerase® is the first biosimilar of dornase alfa, developed by the lead Russian biotechnology company GENERIUM. The aim of the manuscript to present post hoc sub-analysis of patients' data with cystic fibrosis and severe pulmonary impairment of a larger comparative study (phase III open label, prospective, multi-centre, randomized study (NCT04468100)) of a generic version of recombinant human DNase Tigerase® to the only comparable drug, Pulmozyme®

**Data Availability Statement:** All relevant data are within the paper and its Supporting Information file.

**Funding:** The study was initiated and supported by the GENERIUM JSC. GENERIUM JSC had role in the study design, data collection and analysis, decision to publish, and preparation of the manuscript. Elena L. Amelina, Stanislav A. Krasovsky, Nataliya Yu. Kashirskaya, Diana I. Abdulganieva, Irina K. Asherova, Ilya E. Zilber, Liliya S. Kozyreva, Lubov M. Kudelya, Natalya D. Ponomareva, Nataliya P. Revel-Muroz, Elena M. Reutskaya, Tatiana A. Stepanenko, Gulnara N. Seitova, Olga P. Ukhanova, Olga V. Magnitskaya received payment for the above-mentioned clinical trial from the sponsor (GENERIUM JSC).

**Competing interests:** Elena L. Amelina, Stanislav A. Krasovsky, Nataliya Yu. Kashirskaya, Diana I. Abdulganieva, Irina K. Asherova, Ilya E. Zilber, Liliya S. Kozyreva, Lubov M. Kudelya, Natalya D. Ponomareva, Nataliya P. Revel-Muroz, Elena M. Reutskaya, Tatiana A. Stepanenko, Gulnara N. Seitova, Olga P. Ukhanova, Olga V. Magnitskaya received payment for the above-mentioned clinical trial. Dmitry A. Kudlay, Nina E. Akhtyamova-Givirovskaya, Oksana A. Markova, Elena V. Gapchenko are the employees of JSC GENERIUM. This does not alter our adherence to PLOS ONE policies on sharing data and materials.

**Abbreviations:** ADA, anti-drug antibody; AE, adverse event; AR, adverse reaction; CF, cystic fibrosis; FAS, full analyses set; $FEV_1$, forced expiratory volume in 1 second; FVC, forced vital capacity; MedDRA, Medical Dictionary for Regulatory Activities; NCI CTCAE, The National Cancer Institute Common Terminology Criteria for Adverse Events; PP, per protocol; SAE, serious adverse event; W, week.

## Methods

In the analyses included subgroup of 46 severe pulmonary impairment patients with baseline $FEV_1$ level 40–60% of predicted (23 patients in each treatment group) out of 100 patients registered in the study phase III open label, prospective, multi-center, randomized study (NCT04468100), and compared efficacy endpoints ($FEV_1$, FVC, number and time of exacerbations, body weight, St.George's Respiratory Questionnaire) as well as safety parameters (AEs, SAEs, anti-drug antibody) within 24 treatment weeks.

## Results

All outcomes were comparable among the studied groups. In the efficacy dataset, the similar mean $FEV_1$ and mean FVC changes for 24 weeks of both treatment groups were observed. The groups were also comparable in safety, all the secondary efficacy parameters and immunogenicity.

## Conclusions

The findings from this study support the clinical Tigerase® biosimilarity to Pulmozyme® administered in CF patients with severe impairment of pulmonary function.

## 1. Introduction

Cystic fibrosis (CF) is an autosomal recessive disease that affects over 90,000 people worldwide [1]. According to the Research Centre for Medical Genetics and neonatal screening, the frequency of CF in Russia is 1:8,000–12,000 newborns with significant differences by regions [2–5]. The Russian CF Patients Registry, created to assess the prevalence, genetic and clinical polymorphism, approaches to the treatment of the disease, included data of 3,142 patients by 2018 (mean age is 12.8±9.6 years, median age is 10.4 (12.4) years, proportion of patients aged ≥ 18 years is 24.7%) [6, 7]. Since 2011 the Registry patients' number increased from 1,026 in 2011 to 3,142 in 2018. In 2018 171 patients have been diagnosed CF for the first time (162 patients aged < 18 and 9 patients aged ≥18). 124 patients have been diagnosed CF with the neonatal screening, which was 72.9% of all revealed CF cases in 2018. 94.3% of patients were performed genetic analysis. Mean $FEV_1$ (forced expiratory volume in 1 second) and FVC (forced vital capacity) were 77.6 ± 26.1% and 84.6 ± 21.8% of predicted, respectively. The median life expectancy for patients born in 2014–2018 was 33.8 years [6].

CF patients need costly medical care including the comprehensive highly specialized medical services and adequate expensive medicinal products therapy.

Up until recently, only two medicinal products (recombinant human DNase and hypertonic saline) were proved the positive effects on mucous membrane clearance in CF patients [8–11]. Dornase alfa is one of the most used medicinal products in CF patients [12, 13]. According to the Russian CF Patients Registry, in 2018 95.7% of CF patients were administered dornase alfa [6].

The efficacy and safety of dornase alfa have been established through numerous original product Pulmozyme® trials that support its use as an effective treatment option in young CF patients with moderate pulmonary involvement, as well as in adult patients with severe pulmonary impairment. Daily inhalations slow down the pulmonary function impairment and

reduce the frequency of the respiratory disease exacerbations. The most patients tolerate medicinal product well irrespective of the lung damage severity [14–23].

Pulmozyme® is widely used in Russia. Pulmozyme® effective alternative search initiated the domestic equivalent development. Tigerase® biosimilarity (dornase alfa produced by GENERIUM JSC, Russia) to the reference product Pulmozyme® has been proved in the main quality parameters both non-clinical studies and Phase I clinical study in healthy volunteers. The conducted studies displayed good tolerability and a favourable safety profile of the investigational medicinal product [24].

## 2. Material and methods

### 2.1. Study design

The phase III (NCT04468100) clinical trial was aimed to compare the efficacy and safety of Tigerase® vs Pulmozyme® as a treatment component in CF patients.

This phase III open-label, prospective, multi-center, randomized study (NCT04468100) was conducted at 15 clinical sites in the Russian Federation from August 2017 to May 2018 in accordance with the ethical principles of the Helsinki Declaration, and ICH GCP. The approval of the Ministry of Health of Russian Federation and the ethics committee approval were got before recruiting the first participant in a study that meets Russian Federation requirements of clinical trial conducting. Clinical trial approval number of the Ministry of Health of Russia № 348 eff date 23.06.2017. The Ethics Council of the Ministry of Health of the Russian Federation approval dated May 16, 2017. Clinical trial registration on the https:// clinicaltrials.gov is not obligatory requirement in Russian Federation. The trial was registered on the site as completed study (*ClinicalTrials.gov Identifier*: *NCT04468100*). The authors confirm that all ongoing and related trials for this drug are registered.

At all 15 centers the clinical study was approved by the independent ethical committees of healthcare facilities. Each study patient signed an informed consent before undergoing any study procedure.

The manuscript describes post hoc sub-analysis of patients' data with cystic fibrosis and severe pulmonary impairment of the phase III clinical study (NCT04468100) of a generic version of recombinant human DNase Tigerase® to the only comparable drug, Pulmozyme®.

### 2.2. Participants

The clinical study was conducted at 15 clinical centers of the Russian Federation. Patients were only included in the study if they had signed an informed consent form, completed all screening procedures, and met all the inclusion / exclusion criteria. The study included male and female Caucasian patients. Patients of both genders aged 18 and older with a confirmed diagnosis of CF and $FEV_1 \geq 40\%$ of predicted were enrolled. The diagnosis was defined as the presence of the disease clinical evidence along with a positive sweat test and/or detection of 2 clinically significant abnormal *CFTR* gene an exacerbation of the chronic pulmonary disease for 4 weeks prior to and at screening, a condition after lung transplantation, the lung transplantation scheduled for the study period.

During the study, the patients inhaled Tigerase® or reference product Pulmozyme® 2.5 mg once daily with jet nebuliser PARI Compact.

Of the 104 screened patients, totally 100 patients of the main phase III clinical study (NCT04468100) met inclusion/exclusion criteria and were randomly assigned to either control or experimental group with a 1:1 allocation as per a computer-generated randomisation schedule stratified by the $FEV_1$ baseline into two groups (FAS- population): Group I (Tigerase®)– 50 patients and Group II (Pulmozyme®)– 50 patients. Before randomization, the patients

were stratified by the $FEV_1$ baseline: each study group included 23 severe pulmonary impairment ($FEV_1 \geq 40\%$—$\leq 60\%$ of predicted) patients (46.0%) and 27 normal—to—moderate pulmonary function ($FEV_1 > 60\%$—$\leq 100\%$) patients (54.0%). Four patients terminated the study early, 11 patients had significant protocol deviations related to treatment compliance that could affect the study results. Thus, 85 patients (85.0%) completed the study without protocol significant deviations and were included in the per-protocol (PP) population.

The findings only for the patients with severe pulmonary impairment (baseline $FEV_1$ $\geq 40\%$—$\leq 60\%$ of predicted) based on the main randomized clinical trial were analyzed in the manuscript: each study group included 23 severe pulmonary impairment ($FEV_1 \geq 40\%$—$\leq 60\%$ of predicted) patients. The total population study findings were published earlier in the journal of Pulmonology in Russian (2019) [25].

## 2.3. Study endpoints

**2.3.1. Efficacy.** As a primary endpoint the $FEV_1$ change (absolute %) was assessed at study week 24 (W24) compared to the baseline. The secondary efficacy endpoints included the comparison of the following parameters between the treatment groups:

- FVC change (absolute %) at W24 compared to baseline,

- the number of chronic pulmonary disease exacerbations for 24 weeks,

- the time to a chronic pulmonary disease exacerbation over the 24 weeks,

- body weight change at W24 compared to baseline,

- changes in the mean score on the subscales "Symptoms", "Activity", "Impacts", and the mean total score of the St. George's Respiratory Questionnaire, version 2.2, [26] at W24 compared to baseline.

**2.3.2 Safety.** The safety analysis included the following parameters:

- the incidence and severity of adverse events (AEs) and serious adverse events (SAEs) during the study period based on subjective complaints, physical examinations, assessment of vital signs, ECG, laboratory and instrumental tests, and patient diaries;

- the anti-drug antibody (ADA) to dornase alfa formation rate within 24 treatment weeks.

AEs was registered from the time the patient signs an informed consent and until the last visit or procedure related to the study. AE severity has been graded according to The National Cancer Institute Common Terminology Criteria for Adverse Events, (NCI CTCAE), v. 4.03 [27].

## 2.4. Statistical analysis

The main randomized phase III (NCT04468100) clinical trial was planned to test the null hypothesis of the equal efficacy of two medicinal products at W24 versus baseline in the compared groups. The sample size of 100 patients (the number of patients who could have been enrolled based on the feasibility result) was determined to allow to detect the difference ($\delta$) of 5% or more in $\Delta FEV_1$ between the groups with the expected standard deviation ($\sigma$) of 10%, the power of 80% ($Z_\beta = 0.84$) and two-sided $\alpha = 0.05$.

The manuscript reports a post hoc sub-analysis of the group (23 per group) of patients within a larger study had the lowest levels of $FEV_1$.

The trial data were analyzed in three patient populations: 1) all patients included in the study (FAS population, full analyzes set); this population was also used as a baseline in the

analysis of performance parameters; 2) all patients who received at least one dose of the study drug were included in the analysis of safety; the safety population was found to be identical to the FAS population; 3) additionally, the efficacy parameters were analyzed in all patients who completed the study without significant deviations from the Protocol (PP-population, Per protocol). Descriptive statistics are given for each studied parameter. The two-sided t-test was used to analyze between-group differences for the primary efficacy endpoint in the intent-to treat population, in addition, a 95% confidence interval was provided.

The t-test was used for FVC analysis at W24 versus the baseline level. The number of chronic pulmonary disease exacerbations within 24 therapy weeks was analyzed with Poisson regression. Fisher's exact test was used to compare the number and proportion of patients with one, two, three or more exacerbations generally and according to FEV1 baseline stratification between two groups. The time to chronic pulmonary disease exacerbation was analyzed both with the Kaplan–Meier method and the Cox regression model. 'Gender', 'Patient's age', 'Number of concomitant diseases' and 'Treatment group' were considered as covariates. The changes of the mean score on 'Symptoms', 'Activity' and 'Impacts' subscales and mean total score of the Questionnaire at W24 compared to baseline were estimated through paired t-test for intra-group comparison and t-test for group-to-group comparison.

Adverse events were coded by the current Medical Dictionary for Regulatory Activities (MedDRA). Group-to-group comparison of categorical parameters was performed by $\chi^2$-test or Fisher's exact test. The Friedman test was used to assess the quantitative parameters changes at W24 compared to baseline. In the case of statistically significant differences, the post-hoc paired comparison was conducted using a paired t-test or Wilcoxon test depending on the data distribution type. Data on the ADA level of to dornase alfa in both groups were presented through descriptive statistics.

The statistical analysis was performed with the Stata application software (StataCorp, USA) version 14 [28].

## 3. Results

### 3.1. Study population

100 screened patients met the inclusion/exclusion criteria. Of 100 randomized 1:1 into two groups patients, 46 ones (23 patients in each treatment group) composed the severe pulmonary impairment (FEV$_1$ ≥40%—≤60% predicted) strata. 41 patients (89.1%) of the strata completed the study without significant protocol deviations and were included in the PP population (**Fig 1**). Safety population includes all the patients, received at least one dose of study medications.

The study included Caucasian male and female subjects. The treatment groups were comparable in the baseline characteristics, demographic parameters and baseline FEV$_1$ (**Table 1**).

All the enrolled patients received dornase alfa inhalations at a dose of 2.5 mg previously. Before the previous most frequently administered medicinal products for pulmonary disease were bronchodilators, antibiotics and glucocorticosteroids. Also, patients underwent non-medical therapy: kinesitherapy (therapeutic chest massage, physical training, breathing exercises, and postural drainage). Additionally, as part of prior therapy patients took pancreatic enzymes, ursodeoxycholic acid preparations, proton pump inhibitors, anticholinergic agents.

FAS population data analysis has shown that the most common concomitant diseases in both groups were respiratory and gastrointestinal system disorders and infections. The total disorders number in Group I was 178 in 23 (100%) patients, and 192 in 23 (100%) patients in Group II. The various comorbidities incidence was comparable in the studied groups.

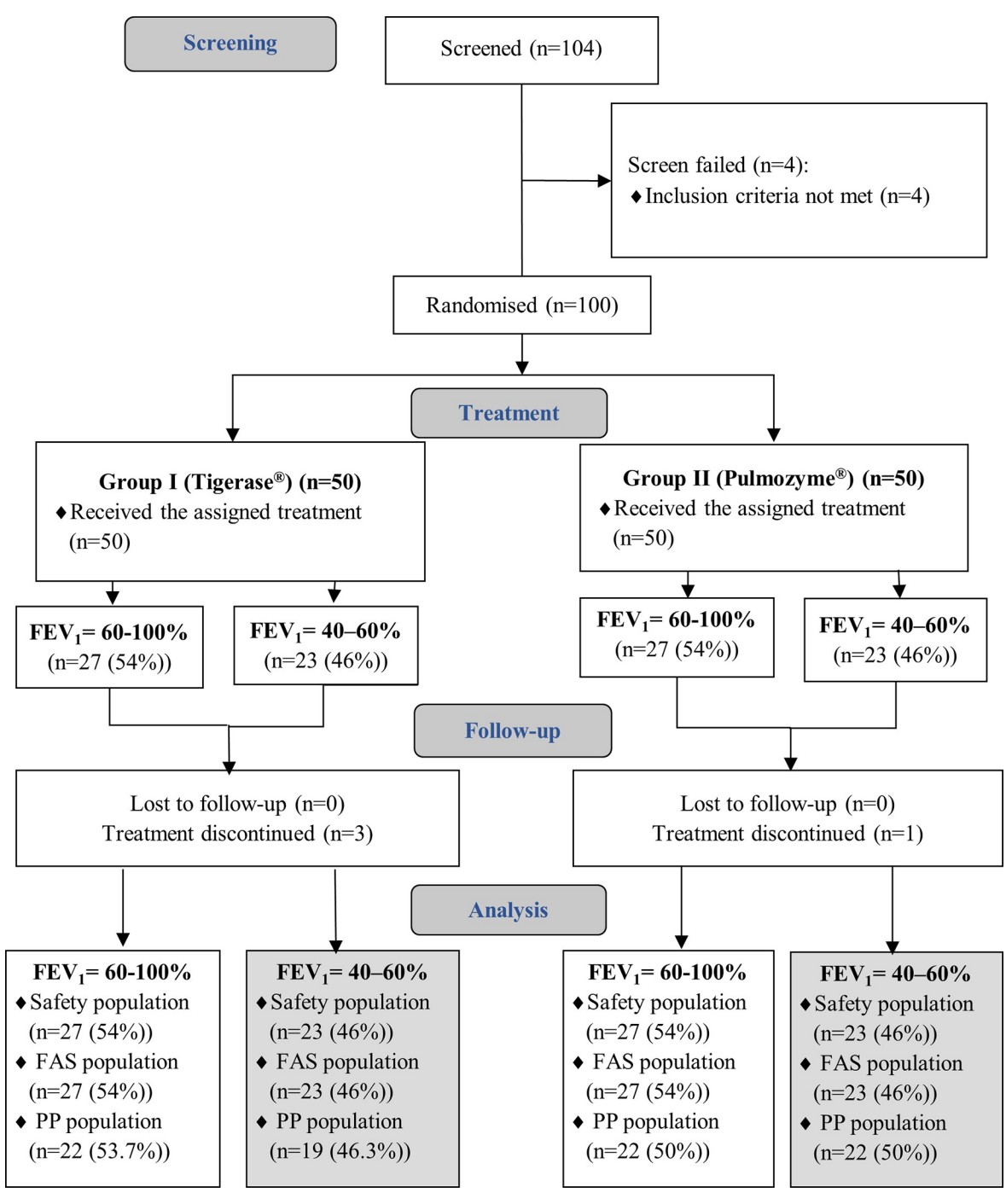

**Fig 1. Patients' distribution.**

At screening the groups were comparable in preexisting immunogenicity profile. The immunogenicity testing revealed preexisting ADA to dornase alfa (IgG) in 1 patient (4.4%) in the Tigerase® group and in 1 patient (4.4%) in the Pulmozyme® group, and IgM ADA were also revealed in 1 patient (4.4%) in each group. Neutralising ADA to dornase alfa were found in 1 patient of Group I, with the titer 5, and in 1 patient of Group II, with the titer 20.

**Table 1. The results of demographic parameters and baseline FEV$_1$.**

| Parameters | | FAS population | | | PP population | | |
|---|---|---|---|---|---|---|---|
| | | Group I (Tigerase®) N = 23 | Group II (Pulmozyme®) N = 23 | p, t-test | Group I (Tigerase®) N = 19 | Group II (Pulmozyme®) N = 22 | p, t-test |
| Age, years, M(SD) | | 30.1 (9.3) | 25.6 (8.8) | 0.098 | 29.7 (8.9) | 25.6 (9.0) | 0.147 |
| Males, n (%) | | 10/23 (43.5) | 12/23 (52.2) | 0.349 | 9/19 (47.4) | 12/22 (54.6) | 0.210 |
| Body weight, kg, M(SD) | | 52.4 (8.4) | 55.8 (10.9) | 0.237 | 52.5 (8.4) | 56.1 (11.1) | 0.257 |
| Height, cm, M(SD) | | 165.0 (8.9) | 166.8 (8.6) | 0.483 | 166.3 (8.8) | 166.8 (8.8) | 0.841 |
| BMI, kg/m$^2$, M(SD) | | 19.18 (2.42) | 20.12 (4.09) | 0.346 | 18.93 (2.31) | 20.22 (4.16) | 0.235 |
| Baseline FEV$_1$, abs. %, M(SD) | 40–60% | 49.6 (5.9) | 50.1 (6.4) | 0.782 | 49.6 (5.7) | 50.3 (6.4) | 0.714 |

No clinically significant abnormalities in physical examinations and electrocardiogram (ECG) were found in both groups at Screening.

Throughout the study, a high level of treatment compliance assessed during the patient's interview at the planned visits, as well as according to the patient's diary was noted. By the middle of the study, the treatment compliance level reached 97.8±6.1% in Group I (Tigerase®) and 98.5±4.0% in Group II (Pulmozyme®) and remained high until the final visit: 94.6±19.1% and 97.8±4.2%, respectively. No statistically significant differences between the groups were found at any of the visits (p>0.05).

### 3.2. Efficacy outcomes

**3.2.1. FEV$_1$ and FVC changes.** Both treatment groups displayed similar changes in the mean FEV$_1$ values at W24 compared to the baseline: the FAS population mean FEV$_1$ value was 48.8±7.4 abs. % (95% CI (45.56; 52.08)) and 48.4±10.0 abs. % (95% CI (44.12; 52.77)) in Group I and II respectively. The findings do not allow the null hypothesis of the equal efficacy of Tigerase® and Pulmozyme® in terms of the FEV$_1$ change during 24 therapy weeks in severe impairment of pulmonary function patients to be rejected.

The PP population similar FEV$_1$ values assessment at W24 also showed comparability between the treatment groups on this parameter (Table 2).

**Table 2. FEV$_1$ and FVC change over time in the study groups (FAS and PP populations).**

| Parameter | Indicator | FAS population | | | PP population | | |
|---|---|---|---|---|---|---|---|
| | | Group I (Tigerase®) N = 23 | Group II (Pulmozyme®) N = 23 | p, t-test | Group I (Tigerase®) N = 19 | Group II (Pulmozyme®) N = 22 | p, t-test |
| FEV$_1$ | Baseline, L (M (95% CI)) | 1.7 (1.52; 1.85) | 1.8 (1.63; 2.00) | 0.281 | 1.7 (1.56; 1.88) | 1.8 (1.64; 2.02) | 0.394 |
| | Baseline, abs. % (M (95% CI)) | 49.6 (47.07; 52.15) | 50.1 (47.36; 52.87) | 0.782 | 49.6 (46.88; 52.37) | 50.3 (47.48; 53.18) | 0.714 |
| | W24, L (M (95% CI)) | 1.7 (1.48; 1.81) | 1.7 (1.53; 1.93) | 0.488 | 1.7 (1.50; 1.84) | 1.7 (1.52; 1.93) | 0.686 |
| | W24, abs. % (M (95% CI)) | 48.8 (45.56; 52.08) | 48.4 (44.12; 52.77) | 0.886 | 48.2 (44.70; 51.74) | 48.0 (43.58; 52.46) | 0.942 |
| | Change in FEV$_1$, abs. % (M (95% CI)) | -0.3 (-2.57; 1.89) | -1.7 (-5.86; 2.51) | 0.566 | -1.4 (-3.52; 0.71) | -2.3 (-6.48; 1.85) | 0.701 |
| FVC | Baseline, L (M (95% CI)) | 3.0 (2.69; 3.36) | 3.1 (2.72; 3.5) | 0.753 | 3.1 (2.69; 3.45) | 3.1 (2.70; 3.51) | 0.905 |
| | Baseline, abs. % (M (95% CI)) | 76.8 (70.46; 83.22) | 73.9 (66.9; 80.82) | 0.516 | 75.9 (68.34; 83.48) | 73.4 (66.18; 80.66) | 0.623 |
| | W24, L (M (95% CI)) | 2.9 (2.58; 3.30) | 2.9 (2.59; 3.28) | 0.979 | 3.0 (2.58; 3.40) | 2.9 (2.55; 3.27) | 0.760 |
| | W24, abs. % (M (95% CI)) | 74.7 (68.14; 81.23) | 70.1 (63.77; 76.44) | 0.303 | 73.5 (66.18; 80.91) | 69.2 (62.85; 75.52) | 0.351 |
| | Change in FVC, abs. % (M (95% CI)) | -1.6 (-5.81; 2.67) | -3.8 (-9.14; 1.64) | 0.515 | -2.4 (-7.22; 2.48) | -4.2 (-9.79; 1.32) | 0.606 |

**Table 3. Poisson regression analysis for the data on the exacerbations of chronic pulmonary disease number for 24 weeks (FAS and PP populations).**

| Covariate | FAS population | | | PP population | | |
|---|---|---|---|---|---|---|
| | IRR | | p | IRR | | p |
| | Point estimate | 95% CI | | Point estimate | 95% CI | |
| Group (Tigerase® / Pulmozyme®) | 0.73 | (0.27; 1.99) | 0.534 | 0.80 | (0.29; 2.18) | 0.663 |
| Gender (male/female) | 0.57 | (0.20; 1.58) | 0.277 | 0.52 | (0.19; 1.43) | 0.205 |
| Age (per 5 years) | 0.94 | (0.71; 1.25) | 0.678 | 0.96 | (0.73; 1.27) | 0.796 |
| Number of concomitant disorders (per 1 disease) | 1.12 | (0.99; 1.27) | 0.061 | 1.11 | (0.98; 1.25) | 0.108 |

No statistically significant effect of the therapy factor on the exacerbation rate was found (p = 0.534 for the 'Group' factor). There was also a tendency of association the concomitant disorders number with the exacerbation risk (p = 0.061 for the 'Number of concomitant disorders' factor).

The between-group differences in $FEV_1$ changes at W24 vs. baseline was 1.4 abs.% (95% CI (-3.3; 6.0), p = 0.566 in the FAS population and 0.9 abs.% (95% CI (-3.8; 5.7)), p = 0.701 in the PP population. Thus, in both populations the between-group differences for $FEV_1$ change did not exceed 6%.

No statistically significant between-group difference in the mean FVC values at W24 was revealed in the FAS and PP populations. The FAS population mean FVC level was 74.7±14.8 abs. % (95% CI (68.14; 81.23)) and 70.1±14.7 abs. % (95% of CI (63.77; 76.44)), p = 0.303 in Group I and II respectively; in the PP population– 73.5±15.3 abs. % (95% of CI (66.18; 80.91)) and 69.2±14.3 abs. % (95% of CI (62.85; 75.52)), p = 0.351 in Group I and II respectively.

Thereby the FVC changes analysis (absolute %) at W24 vs. baseline in the FAS and PP populations revealed no statistically significant between-group differences (Table 2). The mean FVC change value in the FAS population at Week 24±1 was -1.6±9.6% in Group I and -3.8 ±12.5% in Group II. The between-group differences in the FVC change (Group I–Group II) at W24 vs. baseline was 2.2% (95% CI (-4.5; 8.9)), p = 0.515. The mean FVC changes value in the PP population by W24 was -2.4±10.1% in Group I and -4.2±12.5% in Group II. The between-group differences at W24 vs. baseline was 1.9% (95% CI (-5.4; 9.1)), p = 0.606.

**3.2.2. Chronic pulmonary disease exacerbations.** The chronic pulmonary disease exacerbations number during 24 therapy weeks was assessed by the Poisson's regression with the included in the model following covariates: 'Group' (Tigerase® / Pulmozyme®), 'Patient's gender' (male/female), 'Age', 'Number of concomitant diseases'. The chronic pulmonary disease exacerbations incidence rate ratio (IRR) is given in the Table 3.

Group-to-group comparison of the patients' proportion with a different number of chronic pulmonary disease exacerbations for 24 weeks showed no statistically significant differences in both FAS and PP populations (Table 4).

**3.2.3. Time to a chronic pulmonary disease exacerbation.** Comparison of Kaplan–Meier curves through the log-rank test revealed no statistically significant between-group differences

**Table 4. The patients amount with different numbers of chronic pulmonary disease exacerbations for 24±1 weeks (FAS and PP populations).**

| No. of exacerbations | FAS population | | | PP population | | |
|---|---|---|---|---|---|---|
| | Group I (Tigerase®) N = 23 | Group II (Pulmozyme®) N = 23 | p, Fisher test | Group I (Tigerase®) N = 19 | Group II (Pulmozyme®) N = 22 | p, Fisher test |
| 0 | 16/23 (69.6%) | 14/23 (60.9%) | 0.313 | 12/19 (63.2%) | 13/22 (59.1%) | 0.334 |
| 1 | 7/23 (30.4%) | 6/23 (26.1%) | | 7/19 (36.8%) | 6/22 (27.3%) | |
| 2 | 0/23 (0.0%) | 3/23 (13.0%) | | 0/19 (0.0%) | 3/22 (13.6%) | |
| 3 or more | 0/23 (0.0%) | 0/23 (0.0%) | | 0/19 (0.0%) | 0/22 (0.0%) | |

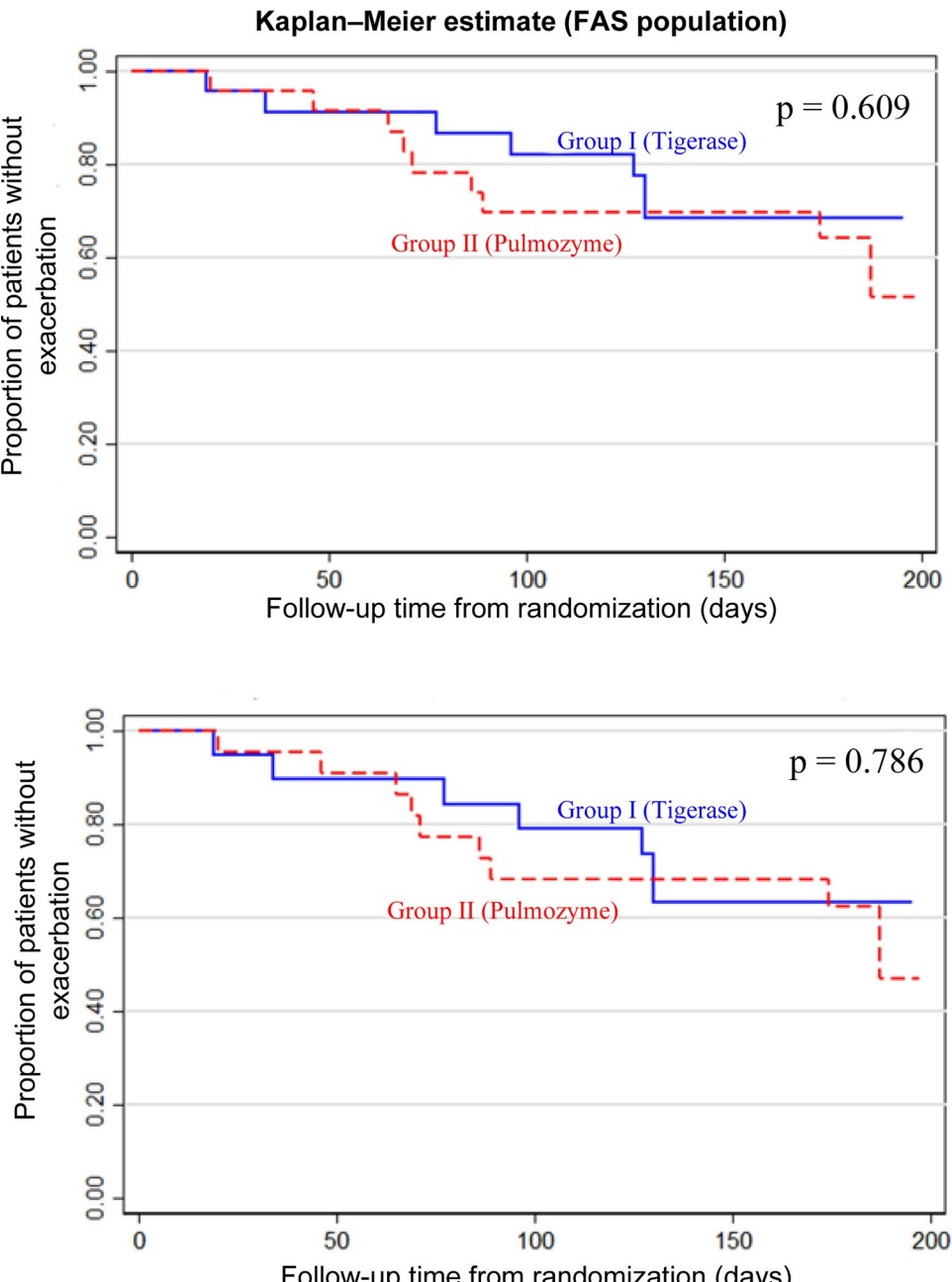

**Fig 2. Kaplan–Meier curves for the time to the first chronic pulmonary disease exacerbation (FAS and PP populations).**

in terms of time to the first chronic pulmonary disease exacerbation for the both FAS (p = 0.609) and PP (p = 0.786) populations (**Fig 2**).

The data analysis of the time to the first chronic pulmonary disease exacerbation through Cox regression showed 'Drug Group', 'Gender' or 'Age' as a non-statistically significant factor for the model (p>0.05). Meanwhile the 'Number of concomitant disorders' factor was statistically significant for the both FAS (p = 0.002) and PP (p = 0.007) population. Hazard ratio for the number of concomitant disorders was 1.27 (95% CI (1.09; 1.48)) in the FAS population (p = 0.002), and 1.23 (95% CI (1.06; 1.44)) in the PP population (p = 0.007).

**Table 5. The mean score changes of the St. George's Respiratory Questionnaire (version 2.2) at W24 vs. baseline data (FAS and PP populations).**

| Subscale | FAS population | | | PP population | | |
|---|---|---|---|---|---|---|
| | Group I (Tigerase®) N = 22 | Group II (Pulmozyme®) N = 23 | p, t-test | Group I (Tigerase®) N = 19 | Group II (Pulmozyme®) N = 22 | p, t-test |
| Symptoms, M (95% CI) | -5.74 (-10.64; -0.85) | -3.30 (-7.47; 0.87) | 0.433 | -5.93 (-11.63; -0.22) | -3.31 (-7.69; 1.06) | 0.446 |
| Activity, M (95% CI) | 2.63 (-4.52; 9.77) | -2.94 (-11.88; 5.99) | 0.320 | 3.40 (-4.84; 11.64) | -3.06 (-12.43; 6.31) | 0.294 |
| Impacts, M (95% CI) | -0.05 (-6.22; 6.12) | 2.09 (-1.71; 5.89) | 0.538 | -1.64 (-8.40; 5.11) | 2.25 (-1.73; 6.22) | 0.290 |
| Total score, M (95% CI) | -0.17 (-5.52; 5.19) | -0.27 (-4.54; 4.00) | 0.975 | -0.81 (-6.92; 5.31) | -0.22 (-4.70; 4.26) | 0.871 |

**3.2.4. Body weight changes.** The body weight analysis demonstrated the similar changes in mean body weight at W24 in both treatment groups: 1.3±2.5 kg (95% CI (0.18; 2.44)) vs. 0.4 ±1.8 kg (95% CI (-0.42; 1.15)) in FAS population (p = 0.159) and 1.5±2.7 kg (95% CI (0.22; 2.81)) vs. 0.4±1.9 kg (95% CI (-0.44; 1.21)) in PP population (p = 0.121), in Group I and II respectively.

Between-group differences in the body weight changes at W24 vs. baseline was 0.9 kg (95% CI -0.4; 2.3) for the FAS population (p = 0.159) and 1.1 kg (95% CI -0.3; 2.6) for the PP population (p = 0.121).

**3.2.5. St. George's respiratory questionnaire.** The St. George's Respiratory Questionnaire scoring showed no statistically significant differences in the total score and in the subscales 'Symptoms', 'Activity', 'Impacts' in both FAS and PP population (**Table 5**).

## 3.3. Safety outcomes

46 patients of the two study groups represented the safety dataset. The medicinal product Tigerase® founded the safety profile similar to Pulmozyme®.

**3.3.1. Incidence of AEs.** In total, the subgroup of severe pulmonary disfunction patients experienced 126 AEs in 34 patients (73.9%): 64 AEs (50.8%) in 15 patients (65.2%) of Tigerase® group and 62 AEs (49.2%) in 19 patients (85.6%) of Pulmozyme® group. The AE rate comparison through Fisher's exact test showed not statistically significant between-group differences.

The most common AEs were infections and infestations (11 patients (47.8%) in each group), investigations (8 patients (34.8%) in Group I; 11 patients (47.8%) in Group II), respiratory, thoracic, and mediastinal disorders (9 patients (39.1%) in Group I; 7 patients (30.4%) in Group II), gastrointestinal disorders (3 patients (12%) in Group I; 2 patients (8.7%) in Group II).

Six AEs in 6 patients (3 in each group) were the maximum AE severity (grade 3), 36 AEs were grade 2 and 84 –grade 1.

Of 126 registered AEs, 92 (73.1%) were resolved with recovery. By the end of the study 8 AEs (6.4%) were resolved with consequences, 23 AEs (18.3%) were not resolved and the outcome of 3 AEs (2.4%) was unknown.

**3.3.2. Incidence of Ars.** Overall, 4 patients experienced 6 adverse reactions (ARs) with 'definite' or 'possible' relationship with the study product: 2 patients in Group I (8,8%)—one with $FEV_1$ reduction ('possible' relationship, grade 2) and one with dysphonia ('definite' relationship, grade 1); and 2 patients in Group II (8,8%)–one with sputum increase, which can be considered as a positive therapy effect ('definite' relationship, grade 1), one patient with 3 AR (shortness of breath, itchy throat, dysphonia), in which the relationship was 'definite', grade 1. No statistically significant between-group differences in patients' number or ARs number

were observed. All ARs were resolved with recovery, except for the $FEV_1$ reduction, which was not resolved by the end of the study.

**3.3.3. Incidence of SAEs.** Five patients experienced 5 serious adverse events (SAEs) (3 patients in Group I and 2 patients in Group II) that required hospitalization or its prolongation. The following SAEs were in Group I: a pulmonary disease exacerbation in 1 patient (4.4%), a pneumothorax in 1 patient (4.4%) and a pneumonia in 1 patient (4.4%). 2 patients (8.7%) in Group II experienced severe pulmonary disease exacerbation. All the SAEs were resolved with recovery.

**3.3.4. Immunogenicity.** During the study, new IgG ADA formation was detected in 4 patients (17,4%) in Group I and in 3 patients (13,1%) in Group II, new formation of IgM ADA were detected in 5 patients (21,7%) in Group I and in 2 patients (8,7%) in Group II. The neutralizing activity of detected ADA was revealed in one patient in each group without any clinical impact.

**3.3.5. Fatality.** No deaths were reported in the study.

## 4. Discussion

This manuscript describes the post hoc sub-analysis of a larger comparative study of a generic version of recombinant human DNase Tigerase® (Generium JSC, Russia) to the current market dominant and only comparable product Pulmozyme® (F. Hoffmann-La Roche Ltd., Switzerland). The full randomized trial (*ClinicalTrials.gov Identifier*: *NCT04468100*) recruited around 100 patients [25]. The full trial protocol can be accessed https://clinicaltrials.gov/ct2/show/NCT04468100.

The primary analysis for the full randomized trial was based population of 100. In the manuscript, we present the analyses on the subpopulation of 46 patients having $FEV_1$ 40–60%. Considering the small number of subgroups patients, we have checked the assumptions, they were met both for full population and subpopulation, no serious violations have been identified. In general, the estimations of the model could be unstable due to the small sample size. However, as we also used the same approach on the sample of 100, we are more confident in the results obtained. Still, rather a small sample size to be considered as one of the limitations of these analyses.

The results of the efficacy and safety of dornase alpha only in the subgroup of 46 patients with cystic fibrosis and severe pulmonary impairment with baseline $FEV_1$ level 40–60% of predicted have not been published in an earlier publication. We suppose the findings of CF patients with severe pulmonary dysfunction (a baseline $FEV_1$ level ≥40%—≤60% predicted) might be more demonstrative and as well confirm the comparability of the proposed biosimilar Tigerase® (Generium JSC, Russia) to the reference Pulmozyme® (F. Hoffmann-La Roche Ltd., Switzerland) taken as part of combined therapy as the results of the full randomized trial.

The equivalence in efficacy between two dornase alfa products was demonstrated based on the $FEV_1$ changes at W24 vs. baseline. $FEV_1$ is considered to be an appropriate outcome for CF studies since low $FEV_1$ values are strongly associated with increased mortality, decreased quality of life [29, 30] as well as greater risk of pulmonary exacerbation and hospitalizations [31].

According to the American Thoracic Society/European Respiratory Society the within-subject changes of $FEV_1$ for normal subjects are within 12% (relative change) in less than one year term [32]. The meaningful changes in $FEV_1$ from 5 or 10% have been reported in CF trials with about 5% in dornase alfa trials. These thresholds are well within the inherent variability of the test [33]. Thus, the detected differences in $FEV_1$ changes between groups of 1.4 abs.% (95% CI (-3.3; 6.0), and 0.9 abs.% (95% CI (-3.8; 5.7)) in the FAS and PP populations are well within the intra-subject variability.

 

All the secondary efficacy endpoints were also comparable among the Tigerase® and Pulmozyme® groups. The chronic pulmonary disease exacerbations rate during 24±1 therapy weeks were not dependent on the treatment group or any other factors except the 'Number of concomitant disorders' which tends to impact and reflects the CF patient's comorbidity. The 'Number of concomitant disorders' factor was also statistically significant for the time to the first exacerbation of chronic pulmonary disease in both treatment groups.

The inclusion of the covariates such as patient's gender, age, number of concomitant diseases in the analysis model was as priory before database lock. The rationale for the inclusion was to reveal the impact of different covariates on the dependent variable and to estimate the adjusted effect of the primary interest.

In both groups all outcomes were comparable among the FAS and PP populations, which indicates the reliability of the study conclusions regarding alternative patient populations selected for the analysis and the sensitivity of the efficacy assessment of the medicinal products to the assumptions underlying the FAS and PP populations. The study results are consistent with the literature data showing dornase alfa efficacy in patients with progressive lung disease. Thus, according to McCoy K. et al. (1996) a 12-week study of the inhaled drug dornase alfa in a group of 320 CF patients with progressive lung disease ($FEV_1 < 40\%$) showed that patients administered dornase alfa had statistically greater $FEV_1$ improvement than the placebo group, with a tendency to decrease general hospitalisation rate [16].

Tigerase® was well tolerated. The safety profile of the proposed biosimilar was favorable and comparable to the reference product. Only AEs such as dysphonia, shortness of breath, itching in the throat, increased sputum were associated with the medicinal product intake and were identical to those reported in patients receiving placebo in a controlled trial of the reference drug Pulmozyme® [34–38].

Both Tigerase® and Pilmozyme® showed low immunogenicity. The newly neutralizing ADA formation to dornase alfa was found in low titer in one patient in Tigerase® group, while the neutralizing ADA were revealed at the Screening Visit and remained throughout the study in one patient received Pulmozyme®. Considering that dornase alfa is used in inhalations, the possibility of a significant decrease in the activity of the medicinal product when neutralising antibodies are detected in plasma is doubtful. The systemic effects of inhaled dornase alfa are known to be negligible [22, 23, 34]. In Phase I study of Tigerase® vs Pulmozyme® were observed only the very low serum concentrations of DNAase [23].

## 5. Conclusion

In conclusion, the data from this randomized, open, prospective multicenter clinical study correspond to the aim that allows the researchers to fulfill their tasks: administration of Tigerase® (JSC Generium, Russia) as part of a combined long-term therapy of CF patients with severe pulmonary dysfunction is comparable to Pulmozyme® (F. Hoffmann-La Roche Ltd., Switzerland) in terms of the efficacy and safety parameters studied.

## Supporting information

**S1 Checklist. CONSORT 2010 checklist of information to include when reporting a randomised trial**\*.
(DOC)

**S1 Dataset.**
(RAR)

                                                                    

**S1 File. Protocol.**
(PDF)

## Author Contributions

**Conceptualization:** Elena L. Amelina, Nina E. Akhtyamova-Givirovskaya, Oksana A. Markova.

**Investigation:** Elena L. Amelina, Stanislav A. Krasovsky, Diana I. Abdulganieva, Irina K. Asherova, Ilya E. Zilber, Liliya S. Kozyreva, Lubov M. Kudelya, Natalya D. Ponomareva, Nataliya P. Revel-Muroz, Elena M. Reutskaya, Tatiana A. Stepanenko, Gulnara N. Seitova, Olga P. Ukhanova, Olga V. Magnitskaya.

**Methodology:** Elena L. Amelina, Stanislav A. Krasovsky, Nataliya Yu. Kashirskaya, Oksana A. Markova.

**Project administration:** Elena V. Gapchenko.

**Resources:** Elena L. Amelina, Stanislav A. Krasovsky, Diana I. Abdulganieva, Irina K. Asherova, Ilya E. Zilber, Liliya S. Kozyreva, Lubov M. Kudelya, Natalya D. Ponomareva, Nataliya P. Revel-Muroz, Elena M. Reutskaya, Tatiana A. Stepanenko, Gulnara N. Seitova, Olga P. Ukhanova, Olga V. Magnitskaya.

**Supervision:** Dmitry A. Kudlay.

**Writing – original draft:** Nina E. Akhtyamova-Givirovskaya.

**Writing – review & editing:** Elena L. Amelina, Nina E. Akhtyamova-Givirovskaya, Nataliya Yu. Kashirskaya, Dmitry A. Kudlay, Oksana A. Markova.

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
