## [Decision Letter · Decision Letter 0]

30 Jun 2021

PONE-D-21-10573

COMPARISON OF BIOSIMILAR TIGERASE® AND PULMOZYME® IN LONG-TERM SYMPTOMATIC THERAPY OF PATIENTS WITH CYSTIC FIBROSIS AND SEVERE PULMONARY IMPAIRMENT

PLOS ONE

Dear Dr. Akhtyamova-Givirovskaya,

Thank you for submitting your manuscript to PLOS ONE. After careful consideration, we feel that it has merit but does not fully meet PLOS ONE’s publication criteria as it currently stands. Therefore, we invite you to submit a revised version of the manuscript that addresses the points raised during the review process.

Please make corrections as suggested by the reviewers or write a detailed rebuttal as a point by point answers and marked changes in the text.

We look forward to receiving your revised manuscript.

Kind regards,

Davor Plavec, MD, MSc, PhD, Prof.

Academic Editor

PLOS ONE

2. Thank you for submitting your clinical trial to PLOS ONE and for providing the name of the registry and the registration number. The information in the registry entry suggests that your trial was registered after patient recruitment began. PLOS ONE strongly encourages authors to register all trials before recruiting the first participant in a study.

a) your reasons for your delay in registering this study (after enrolment of participants started)

b) confirmation that all related trials are registered by stating: “The authors confirm that all ongoing and related trials for this drug/intervention are registered”.

3. Thank you for including your ethics statement:  "This phase III open-label, prospective, multi-center, randomized study was conducted at 15 clinical sites in the Russian Federation from August 2017 to May 2018 in accordance with the ethical principles of the Helsinki Declaration, and ICH GCP. At all 15 centers the clinical study was approved by the independent ethical committees of healthcare facilities. Each study patient signed an informed consent before undergoing any study procedure. Clinical trial approval number of the Ministry of Health of Russia № 348 eff date 23.06.2017

The ethics committee approval before the trial began can be accessed .  " ext-link-type="uri" xlink:type="simple">https://clck.ru/U7AEm".  

4. In your Methods section, please provide additional information about the participant recruitment method and the demographic details of your participants. Please ensure you have provided sufficient details to replicate the analyses such as: aa a description of how participants were recruited, and b) descriptions of where participants were recruited and where the research took place.

5. Please ensure you have discussed any potential limitations of your study in the Discussion, including study design, sample size and/or potential confounders.

6. Thank you for stating the following financial disclosure:

“This study was sponsored by GENERIUM JSC. The decision to submit the manuscript was made by the authors and approved by GENERIUM JSC. All listed authors approved the article for submission.”

7. Thank you for stating the following in the Competing Interests section:

“Elena L. Amelina, Stanislav A. Krasovsky, Nataliya Yu. Kashirskaya, Diana I. Abdulganieva, Irina K. Asherova, Ilya E. Zilber, Liliya S. Kozyreva, Lubov M. Kudelya, Natalya D. Ponomareva, Nataliya P. Revel-Muroz, Elena M. Reutskaya, Tatiana A. Stepanenko, Gulnara N. Seitova, Olga P. Ukhanova, Olga V. Magnitskaya received payment for the above mentioned clinical trial. Dmitry A. Kudlay, Nina E. Akhtyamova-Givirovskaya, Oksana A. Markova, Elena V. Gapchenko are the employees of JSC GENERIUM.”

Additional Editor Comments (if provided):

Dear Authors,

please make corrections to your manuscript according to suggestions and comments made by the reviewers or write a detailed rebuttal.

Reviewers' comments:

Reviewer's Responses to Questions

**Comments to the Author**

1. Is the manuscript technically sound, and do the data support the conclusions?

Reviewer #1: Yes

Reviewer #2: Partly

Reviewer #3: Yes

2. Has the statistical analysis been performed appropriately and rigorously? 

Reviewer #1: Yes

Reviewer #2: Yes

Reviewer #3: Yes

3. Have the authors made all data underlying the findings in their manuscript fully available?

Reviewer #1: Yes

Reviewer #2: Yes

Reviewer #3: Yes

4. Is the manuscript presented in an intelligible fashion and written in standard English?

Reviewer #1: Yes

Reviewer #2: No

Reviewer #3: Yes

5. Review Comments to the Author

Reviewer #1: The paper is technically sound, and the conclusions are in accordance with the presented data. The statistical analysis was done correctly. The authors make all data available completely without restriction. The topic of the paper is addressed in a comprehensible way and in standard English.

I have some minor objections:

- I am not sure if “The St. George's Respiratory Questionnaire”, a respiratory disease specific questionnaire, is optimal for assessing the quality of life of patients with cystic fibrosis in trials involving the effect of the drug only on lung disease, considering that CF is not just lung disease.

Please could the authors look into the details stated below and make the needed corrections:

- Page 8:

NCI CTCAE - The National Cancer 175 Institute Common Terminology Criteria for Adverse Events - is not stated in the list of abbreviation

- Page 16:

329 3.3.2. Incidence of ADRs – ADR is not stated in the list of abbreviations, only AR. This should be synchronized.

- Page 20:

404 Tigerase® or Tigeraza® What is correct?

Reviewer #2: This is a very interesting studying evaluating the efficacy of dornase alfa in people with cystic fibrosis. In a subgroup of people from the main randomised controlled trial.

The major comment is the manuscript would benefit from the some restructuring as it is hard to follow-up the various section. Its only when you keep reading when you realize that this is a subgroup analyses.

They are some comments worth mentioning for the authors attention.

1) Can the ‘background’ in the abstract include the main objective of the study.

2) Can the title indicate that this is subgroup analysis, (see comment number 5 below) of a Phase III randomised open-label trial. Clarity is needed.

3) The authors should restructure the methods section, so that its easier to follow-up. For example, having a sub heading of subgroup analysis under materials and heading is very confusing as this this should come after the main analysis section. Although it looks like the subgroup analysis relates to inclusion/exclusion criteria, this should be made clear.

4) Also details relating on registration should really be in the methods section not the introduction section. i.e line 124.

5) Line 152/182 suggests this is a subgroup analysis study, based on a main RCT. This again needs to made clear in the intro objective and methods section.

6) Line 177 “The study” assuming this is based on the main RCT, this again should be made clear.

7) Is line 142 to 151 about the main study and in relation to the subgroup population used in this analysis?

8) Under the statistical analysis section with such few patients and the models fitted, were the assumption checked?

9) What was the rationale for the inclusion of gender, patients (is this age at baseline?), number of concomitant disease in the model, was this decided as priory?

10) Different populations have been stated in the manuscript, FAS, ITT PP and safety. It would help the reader if this placed in section with an explicit statement about which population is being considered in the subgroup analysis e.g. line 211 mentions included in the PP. results present results stratified by populations. Planned presentation of results should also be mentioned in the methods section.

11) Perhaps with such few patients in each of the groups descriptive statistics might suffice.

12) In the discussion, any mention of limitations?

13) Figure 2, is the x-axis observation time from randomisation? This should labelled so. And the y-axis the label is not clear “proportion of patients without”?

Reviewer #3: In the manuscript; PONE-D-21-10573

COMPARISON OF BIOSIMILAR TIGERASE® AND PULMOZYME® IN LONG-TERM SYMPTOMATIC THERAPY OF PATIENTS WITH CYSTIC FIBROSIS AND SEVERE PULMONARY IMPAIRMENT, the authors describes post hoc sub-analysis of a larger comparative study (phase III open label, prospective, multi-centre, randomized study) of a generic version of recombinant human DNase Tigerase® to the only comparable drug, Pulmozyme®. In the analyses included subgroup of 46 severe pulmonary impairment patients with baseline FEV1 level 40-60% of predicted (23 patients in each treatment group), and comapred efficacy endpoints (FEV1, FVC, number and time of exacerbations, body weight, St.George's Respiratory Questionnaire) as well as safety parameters (AEs, SAEs, anti-drug antibody) within 24 treatment weeks.

I suggest to the authors the following clarifications and additions;

- state the strengths and weaknesses of this publication in the discussion

- explain why they analyzed the efficacy and safety of dornase alpha only in the group of patients with cystic fibrosis and severe pulmonary impairment with baseline FEV1 level 40-60% of prediced, and whether the results have already been published in an earlier publication (reference 24)?

-explain why the average age of the included subjects was approximately 30 (group I), vs 25.6 years (group II), ie what was the share of patients with delta F508 mutation among them.

-present the rate of exacerbations requiring antibiotics.

6. PLOS authors have the option to publish the peer review history of their article (what does this mean?). If published, this will include your full peer review and any attached files.

Reviewer #1: No

Reviewer #2: No

Reviewer #3: No

---

## [Author Response · Author response to Decision Letter 0]

4 Aug 2021

Dear Editor,

Thank you for your careful consideration of our manuscript PONE-D-21-10573

COMPARISON OF BIOSIMILAR TIGERASE® AND PULMOZYME® IN LONG-TERM SYMPTOMATIC THERAPY OF PATIENTS WITH CYSTIC FIBROSIS AND SEVERE PULMONARY IMPAIRMENT

PLOS ONE. 

Our replies to the editor and the reviewers’ comments are placed below.

Answer: Thank you for your comment! Our manuscript meets PLOS ONE's style requirements, including those for file naming.

2. Thank you for submitting your clinical trial to PLOS ONE and for providing the name of the registry and the registration number. The information in the registry entry suggests that your trial was registered after patient recruitment began. PLOS ONE strongly encourages authors to register all trials before recruiting the first participant in a study.

a) your reasons for your delay in registering this study (after enrolment of participants started)

b) confirmation that all related trials are registered by stating: “The authors confirm that all ongoing and related trials for this drug/intervention are registered”.

Answer: Thank you for your comment! Start date of the study (date of inclusion of the first patient) was 30.08.2017. The approval of the Ministry of Health of Russian Federation and the ethics committee approval were got before the trial began - РКИ №348 (23.06.2017) 

https://grls.rosminzdrav.ru/CiPermitionReg.aspx?PermYear=0DateInc=NumInc=DateBeg=DateEnd=Protocol=DRN-CFR-II%2fIIIRegNm=Statement=ProtoId=idCIStatementCh=Qualifier=CiPhase=RangeOfApp=Torg=%d0%a2%d0%b8%d0%b3%d0%b5%d1%80%d0%b0%d0%b7%d0%b0LFDos=Producer=Recearcher=sponsorCountry=MedBaseCount=CiType=PatientCount=OrgDocOut=2Status=2NotInReg=0All=0PageSize=8order=datepermorderType=descpagenum=1

https://grls.rosminzdrav.ru/CIPermissionMini.aspx?CIStatementGUID=e1aa8df0-7e40-44b9-ac32-0fb37ae29b44CIPermGUID=83FE08A6-E341-4F9F-9F3A-759B61F479BD

This meets Russian Federation requirements of clinical trial conducting. Clinical trial registration on the https://clinicaltrials.gov is not obligatory requirement in Russian Federation. The trial was registered on the site as completed study.

The necessary remarks were included in the Methods section of our paper.

3. Thank you for including your ethics statement: "This phase III open-label, prospective, multi-center, randomized study was conducted at 15 clinical sites in the Russian Federation from August 2017 to May 2018 in accordance with the ethical principles of the Helsinki Declaration, and ICH GCP. At all 15 centers the clinical study was approved by the independent ethical committees of healthcare facilities. Each study patient signed an informed consent before undergoing any study procedure. Clinical trial approval number of the Ministry of Health of Russia № 348 eff date 23.06.2017

The ethics committee approval before the trial began can be accessed https://clck.ru/U7AEm". 

Answer: Thank you for your comment! The approval of the Ministry of Health of Russian Federation and the ethics committee approval were got before recruiting the first participant in a study that meets Russian Federation requirements of clinical trial conducting. Clinical trial approval number of the Ministry of Health of Russia № 348 eff date 23.06.2017. The Ethics Council of the Ministry of Health of the Russian Federation approval dated May 16, 2017.

The necessary remarks were included in the Methods section of the manuscript and added the same text to the “Ethics Statement” field of the submission form (via “Edit Submission”).

4. In your Methods section, please provide additional information about the participant recruitment method and the demographic details of your participants. Please ensure you have provided sufficient details to replicate the analyses such as: aa a description of how participants were recruited, and b) descriptions of where participants were recruited and where the research took place.

Answer: Thank you for your comment! The needed corrections have been made in the Method section.

5. Please ensure you have discussed any potential limitations of your study in the Discussion, including study design, sample size and/or potential confounders.

Answer: Thank you for your comment! Any potential limitations of our study have been discussed in the Discussion of the manuscript.

6. Thank you for stating the following financial disclosure:

“This study was sponsored by GENERIUM JSC. The decision to submit the manuscript was made by the authors and approved by GENERIUM JSC. All listed authors approved the article for submission.”

Answer: Thank you for your comment! The study was initiated and supported by the GENERIUM JSC. 

GENERIUM JSC had role in the study design, data collection and analysis, decision to publish, and preparation of the manuscript.

Elena L. Amelina, Stanislav A. Krasovsky, Nataliya Yu. Kashirskaya, Diana I. Abdulganieva, Irina K. Asherova, Ilya E. Zilber, Liliya S. Kozyreva, Lubov M. Kudelya, Natalya D. Ponomareva, Nataliya P. Revel-Muroz, Elena M. Reutskaya, Tatiana A. Stepanenko, Gulnara N. Seitova, Olga P. Ukhanova, Olga V. Magnitskaya received payment for the above-mentioned clinical trial from the sponsor (GENERIUM JSC).

7. Thank you for stating the following in the Competing Interests section:

“Elena L. Amelina, Stanislav A. Krasovsky, Nataliya Yu. Kashirskaya, Diana I. Abdulganieva, Irina K. Asherova, Ilya E. Zilber, Liliya S. Kozyreva, Lubov M. Kudelya, Natalya D. Ponomareva, Nataliya P. Revel-Muroz, Elena M. Reutskaya, Tatiana A. Stepanenko, Gulnara N. Seitova, Olga P. Ukhanova, Olga V. Magnitskaya received payment for the above mentioned clinical trial. Dmitry A. Kudlay, Nina E. Akhtyamova-Givirovskaya, Oksana A. Markova, Elena V. Gapchenko are the employees of JSC GENERIUM.”

Answer: Thank you for your comment! Elena L. Amelina, Stanislav A. Krasovsky, Nataliya Yu. Kashirskaya, Diana I. Abdulganieva, Irina K. Asherova, Ilya E. Zilber, Liliya S. Kozyreva, Lubov M. Kudelya, Natalya D. Ponomareva, Nataliya P. Revel-Muroz, Elena M. Reutskaya, Tatiana A. Stepanenko, Gulnara N. Seitova, Olga P. Ukhanova, Olga V. Magnitskaya received payment for the above-mentioned clinical trial. Dmitry A. Kudlay, Nina E. Akhtyamova-Givirovskaya, Oksana A. Markova, Elena V. Gapchenko are the employees of JSC GENERIUM. This does not alter our adherence to PLOS ONE policies on sharing data and materials.

 

Additional Editor Comments 

Comments to the Author

1. Is the manuscript technically sound, and do the data support the conclusions?

Answer: Thank you for your comment! We confirm that the manuscript is technically sound, and the conclusions are in accordance with the presented data. The statistical analysis was done correctly. All data are available completely without restriction.

2. Has the statistical analysis been performed appropriately and rigorously? 

Answer: Thank you for your comment! We confirm the statistical analysis has been performed appropriately and rigorously.

3. Have the authors made all data underlying the findings in their manuscript fully available?

Answer: Thank you for your comment! All data underlying the findings described in our manuscript are fully available without restriction, the appropriate excel files with the data were sent to the edition of the journal PLOS ONE.

4. Is the manuscript presented in an intelligible fashion and written in standard English?

Answer: Thank you for your comment! The manuscript is written in standard English, has checked and is presented in an intelligible fashion.

5. Review Comments to the Author

 

Reviewer #1: The paper is technically sound, and the conclusions are in accordance with the presented data. The statistical analysis was done correctly. The authors make all data available completely without restriction. The topic of the paper is addressed in a comprehensible way and in standard English.

I have some minor objections:

- I am not sure if “The St. George's Respiratory Questionnaire”, a respiratory disease specific questionnaire, is optimal for assessing the quality of life of patients with cystic fibrosis in trials involving the effect of the drug only on lung disease, considering that CF is not just lung disease.

Answer: Thank you for your comment! We completely agree that CF is not just lung disease. According to the literature data validity and reliability of the St George's Respiratory Questionnaire were proved in adults with cystic fibrosis. Self-perceived quality of life is worse among adults with CF than in the general population or among patients with chronic obstructive pulmonary disease. The SGRQ is a valid instrument for analyzing health-related quality of life in adults with CF as it discriminates very well between different degrees of severity of pulmonary impairment and also have an appropriate intern consistency.

Please could the authors look into the details stated below and make the needed corrections:

- Page 8:

NCI CTCAE - The National Cancer 175 Institute Common Terminology Criteria for Adverse Events - is not stated in the list of abbreviation.

Answer: Thank you for your comment! The needed corrections have been made.

- Page 16:329 3.3.2. Incidence of ADRs – ADR is not stated in the list of abbreviations, only AR. This should be synchronized.

Answer: Thank you for your comment! The needed corrections have been made.

- Page 20:404 Tigerase® or Tigeraza® What is correct?

Answer: Thank you for your comment! Tigerase® is correct. The needed corrections have been made.

 

Reviewer #2: This is a very interesting studying evaluating the efficacy of dornase alfa in people with cystic fibrosis. In a subgroup of people from the main randomised controlled trial.

The major comment is the manuscript would benefit from the some restructuring as it is hard to follow-up the various section. Its only when you keep reading when you realize that this is a subgroup analyses. They are some comments worth mentioning for the authors attention.

1) Can the ‘background’ in the abstract include the main objective of the study.

Answer: Thank you for your comment! The needed corrections have been made.

2) Can the title indicate that this is subgroup analysis, (see comment number 5 below) of a Phase III randomised open-label trial. Clarity is needed.

Answer: Thank you for your comment! The needed corrections have been made.

3) The authors should restructure the methods section, so that its easier to follow-up. For example, having a sub heading of subgroup analysis under materials and heading is very confusing as this this should come after the main analysis section. Although it looks like the subgroup analysis relates to inclusion/exclusion criteria, this should be made clear.

Answer: Thank you for your comment! The needed corrections have been made.

4) Also details relating on registration should really be in the methods section not the introduction section. i.e line 124.

Answer: Thank you for your comment! The needed corrections have been made.

5) Line 152/182 suggests this is a subgroup analysis study, based on a main RCT. This again needs to made clear in the intro objective and methods section.

Answer: Thank you for your comment! The needed corrections have been made. It has been clarified in the methods section (2. Material and methods � 2.2.1.Study design).

6) Line 177 “The study” assuming this is based on the main RCT, this again should be made clear.

Answer: Thank you for your comment! The needed corrections have been made.

7) Is line 142 to 151 about the main study and in relation to the subgroup population used in this analysis?

Answer: Thank you for your comment! The issue has been corrected. Totally 100 patients were included in the main RCT, the findings only for the patients with severe pulmonary impairment (baseline FEV1 ≥40% - ≤60% of predicted) were analyzed in the manuscript.

8) Under the statistical analysis section with such few patients and the models fitted, were the assumption checked?

Answer: the assumptions were checked and no serious violations were revealed. 

9) What was the rationale for the inclusion of gender, patients (is this age at baseline?), number of concomitant disease in the model, was this decided as priory?

Answer: this was defined by the SAP that was finalized before database lock. The rational was this was 1) to reveal an impact of different covariates on the dependent variable, 2) to estimate the adjusted effect of the primary interest.

10) Different populations have been stated in the manuscript, FAS, ITT PP and safety. It would help the reader if this placed in section with an explicit statement about which population is being considered in the subgroup analysis e.g. line 211 mentions included in the PP. results present results stratified by populations. Planned presentation of results should also be mentioned in the methods section.

Answer: Thank you for your comment! The needed corrections have been made in the Method section (2.4. Statistical analysis).

11) Perhaps with such few patients in each of the groups descriptive statistics might suffice.

12) In the discussion, any mention of limitations?

Answer for 11 and 12: The primary analysis for study was based population of 100. In the paper we present the analyses on the subpopulation of patients having FEV1 40-60%, also we have checked the assumptions they were met both for full population and subpopulation. In general, we agree that in the model the estimations could be unstable due to a small sample size. However. as we also used same approach on a sample of 100 we are more confident in the results obtained. Still a rather small sample size to be considered as one of the limitation of these analyses. 

13) Figure 2, is the x-axis observation time from randomization? This should labelled so. And the y-axis the label is not clear “proportion of patients without”?

Answer: Thank you for your comment! The needed corrections have been made: the x-axis is titled; the label of the y-axis is corrected. 

 

Reviewer #3: In the manuscript; PONE-D-21-10573

COMPARISON OF BIOSIMILAR TIGERASE® AND PULMOZYME® IN LONG-TERM SYMPTOMATIC THERAPY OF PATIENTS WITH CYSTIC FIBROSIS AND SEVERE PULMONARY IMPAIRMENT, the authors describes post hoc sub-analysis of a larger comparative study (phase III open label, prospective, multi-centre, randomized study) of a generic version of recombinant human DNase Tigerase® to the only comparable drug, Pulmozyme®. In the analyses included subgroup of 46 severe pulmonary impairment patients with baseline FEV1 level 40-60% of predicted (23 patients in each treatment group), and comapred efficacy endpoints (FEV1, FVC, number and time of exacerbations, body weight, St.George's Respiratory Questionnaire) as well as safety parameters (AEs, SAEs, anti-drug antibody) within 24 treatment weeks.

I suggest to the authors the following clarifications and additions;

- state the strengths and weaknesses of this publication in the discussion

- explain why they analyzed the efficacy and safety of dornase alpha only in the group of patients with cystic fibrosis and severe pulmonary impairment with baseline FEV1 level 40-60% of prediced, and whether the results have already been published in an earlier publication (reference 24)?

-explain why the average age of the included subjects was approximately 30 (group I), vs 25.6 years (group II), ie what was the share of patients with delta F508 mutation among them.

-present the rate of exacerbations requiring antibiotics.

Answer: Thank you for your comment! The needed corrections why the average age was different have been made in the Discussion section. As the study was randomized, the average age disbalance is a random disbalance due to pure chance.

- As for share of patients with delta F508 mutation it was not the aim of the trial, CF diagnosis was defined as the presence of the disease clinical evidence along with a positive sweat test and/or detection of 2 clinically significant abnormal CFTR gene. If a patient met the criteria, he/she was included.

- As for the rate of exacerbations requiring antibiotics there were 19 patients of 46 patients with baseline FEV1 level 40-60% of predicted (7 patients in the group of Tigerase® and 12 patients in the group of Pulmozyme®) with exacerbations all of them had required antibiotics.

6. PLOS authors have the option to publish the peer review history of their article (what does this mean?). If published, this will include your full peer review and any attached files.

Do you want your identity to be public for this peer review? For information about this choice, including consent withdrawal, please see our Privacy Policy.

Answer: Thank you for your comment! We would rather not publish the peer review history of our article. All important discussed issues were added to the manuscript.

Thank you for your consideration of this manuscript.

Sincerely,

Nina E. Akhtyamova-Givirovskaya.

---

## [Decision Letter · Decision Letter 1]

2 Dec 2021

COMPARISON OF BIOSIMILAR TIGERASE® AND PULMOZYME® IN LONG-TERM SYMPTOMATIC THERAPY OF PATIENTS WITH CYSTIC FIBROSIS AND SEVERE PULMONARY IMPAIRMENT (subgroup analysis of a Phase III randomized open-label clinical trial (NCT04468100))

PONE-D-21-10573R1

Dear Dr. Akhtyamova-Givirovskaya,

We’re pleased to inform you that your manuscript has been judged scientifically suitable for publication and will be formally accepted for publication once it meets all outstanding technical requirements.

Kind regards,

Davor Plavec, MD, MSc, PhD, Prof.

Academic Editor

PLOS ONE

Additional Editor Comments (optional):

Based on the decisions of reviewers the manuscript is now ready for publication in its current form.

Reviewers' comments:

Reviewer's Responses to Questions

**Comments to the Author**

1. If the authors have adequately addressed your comments raised in a previous round of review and you feel that this manuscript is now acceptable for publication, you may indicate that here to bypass the “Comments to the Author” section, enter your conflict of interest statement in the “Confidential to Editor” section, and submit your "Accept" recommendation.

Reviewer #1: All comments have been addressed

Reviewer #2: All comments have been addressed

Reviewer #3: All comments have been addressed

2. Is the manuscript technically sound, and do the data support the conclusions?

Reviewer #1: (No Response)

Reviewer #2: Yes

Reviewer #3: Yes

3. Has the statistical analysis been performed appropriately and rigorously? 

Reviewer #1: (No Response)

Reviewer #2: Yes

Reviewer #3: Yes

4. Have the authors made all data underlying the findings in their manuscript fully available?

Reviewer #1: (No Response)

Reviewer #2: No

Reviewer #3: Yes

5. Is the manuscript presented in an intelligible fashion and written in standard English?

Reviewer #1: (No Response)

Reviewer #2: Yes

Reviewer #3: Yes

6. Review Comments to the Author

Reviewer #1: (No Response)

Reviewer #2: (No Response)

Reviewer #3: The authors have adequately addressed my comments raised in a previous round of review and I feel that this manuscript is now acceptable for publication.

7. PLOS authors have the option to publish the peer review history of their article (what does this mean?). If published, this will include your full peer review and any attached files.

Reviewer #1: **Yes: **dr. Dorian Tjesic-Drinkovic

Reviewer #2: No

Reviewer #3: No

---

## [Editor Report · Acceptance letter]

10 Dec 2021

PONE-D-21-10573R1 

COMPARISON OF BIOSIMILAR TIGERASE AND PULMOZYME IN LONG-TERM SYMPTOMATIC THERAPY OF PATIENTS WITH CYSTIC FIBROSIS AND SEVERE PULMONARY IMPAIRMENT (subgroup analysis of a Phase III randomized open-label clinical trial (NCT04468100)) 

Dear Dr. Akhtyamova-Givirovskaya:

I'm pleased to inform you that your manuscript has been deemed suitable for publication in PLOS ONE. Congratulations! Your manuscript is now with our production department. 

Kind regards, 

on behalf of

Dr. Davor Plavec 

Academic Editor

PLOS ONE